# A Comparison of the Antioxidant Potential and Metabolite Analysis of Marine Fungi Associated with the Red Algae *Pterocladiella capillacea* from Northern Taiwan

**DOI:** 10.3390/antiox13030336

**Published:** 2024-03-11

**Authors:** Jiji Kannan, Ka-Lai Pang, Ying-Ning Ho, Pang-Hung Hsu, Li-Li Chen

**Affiliations:** 1Institute of Marine Biology, National Taiwan Ocean University, Keelung 20224, Taiwan; 20934003@mail.ntou.edu.tw (J.K.); klpang@ntou.edu.tw (K.-L.P.); ynho@mail.ntou.edu.tw (Y.-N.H.); 2Department of Bioscience and Biotechnology, National Taiwan Ocean University, Keelung 20224, Taiwan; 3Center of Excellence for the Oceans, National Taiwan Ocean University, Keelung 20224, Taiwan

**Keywords:** marine fungi, antioxidant activity, secondary metabolites, cytotoxicity

## Abstract

This study represents a primary investigation centered on screening six marine fungi, *Emericellopsis maritima*, *Engyodontium album*, *Hypomontagnella monticulosa*, *Hortaea werneckii*, *Trichoderma harzianum*, and *Aspergillus* sp.7, associated with the red algae *Pterocladiella capillacea*, which was collected from Chao-Jin Park in Keelung, Taiwan, as potential immunostimulants for shrimp aquaculture. Recognizing the imperative for novel strategies to combat pathogen resistance arising from the use of antibiotics and vaccines in aquaculture, this study aimed to evaluate the metabolomic profile, antioxidant capabilities, and antibacterial properties of marine fungi. The antibacterial activity of the fungal extract was evaluated against five major aquaculture pathogens: *Bacillus subtilis*, *Escherichia coli*, *Staphylococcus aureus*, *Enterobacter aeruginosa*, and *Vibrio parahaemolyticus*. The viability and cytotoxicity of marine fungal extracts were preliminarily evaluated using brine shrimps before assessing cytotoxicity, growth performance, immune efficacy, and disease resistance in white shrimp. The present study demonstrated that total phytochemical analysis correlated with antioxidant activity. *Emericellopsis maritima* and *Trichoderma harzianum* exhibited the strongest DPPH antioxidant scavenging activities of half-maximal inhibitory concentration (IC50) 16.5 ± 1.2 and 12.2 ± 2.6, which are comparable to ascorbic acid. LC-HDMSE analysis of the marine fungal extracts identified more than 8000 metabolites mainly classified under the superclass level of organic oxygen compounds, Organoheterocyclic compounds, Phenylpropanoids and polyketides, alkaloid and derivatives, benzenoids, lignans and neolignans, lipid and lipid-like molecules, nucleotides and nucleosides, organic nitrogen compounds, and organic acids and derivatives. Overall, our study significantly contributes to the advancement of sustainable practices by exploring alternative antimicrobial solutions and harnessing the bioactive potential inherent in marine endophytic fungi. In conclusion, our study advances our comprehension of fungal communities and their applications and holds promise for the development of effective and environmentally friendly approaches for enhancing shrimp health and productivity.

## 1. Introduction

Over the past two decades, antibiotics and chemotherapeutics have been extensively employed in aquaculture farming to prevent various infections and thereby enhance the survival and growth performance of cultivated species [1]. Farm-raised aquatic organisms experience persistent stress, which raises concerns regarding their gut health and the potential for an imbalance known as dysbiosis. Given the extensive use of antibiotics in aquaculture and the emergence of pathogens, there is an escalating demand for cost-effective, environmentally friendly, and sustainable alternatives for intensive aquaculture species production [2,3]. Recently, there has been a growing recommendation and utilization of beneficial microorganisms, such as bacteria and fungi, as probiotics and prebiotics. Fungi serve as abundant sources of diverse and crucial natural products. In the past decade, considerable attention has been directed towards the polysaccharides synthesized by various fungi [4,5,6]. In a recent study, it was reported that the three bioactive subfractions of low-molecular-weight secondary metabolites, ex-LMSI, ex-LMSII, and ex-LMSIII, isolated from *Cerrena unicolor* culture fluid, possessed anticancer, antioxidant, and antibacterial properties [7]. In recent years, scientific research has experienced a remarkable shift towards microorganisms as promising sources of novel natural products. Microorganisms offer a unique advantage by providing a continuous and sustainable supply of bioactive compounds. This makes them particularly attractive to industries that require significant biomass quantities for various purposes, spanning from clinical testing to industrial production [8,9,10]. Fungi have gained prominence in the field of marine bioprospecting. Their ability to produce an array of bioactive metabolites, including antibiotics, anticancer agents, antioxidants, and antiviral compounds, has garnered significant attention. Moreover, there is a growing emphasis on the quest for natural antioxidant compounds that have the potential to replace their currently utilized synthetic counterparts [11,12,13,14].

To thrive in extreme environments, such as the marine ecosystem, fungi have developed various strategies, one of which is the production of secondary metabolites. These compounds enhance their ability to withstand the challenging conditions prevalent in hypersaline environments [15]. However, there exists a notable research gap concerning the exploration of endophytic fungi’s applications in the pharmaceutical industry [16]. Limited attention has been devoted to understanding the potential benefits and applications of these fungi in the development of innovative pharmaceutical products [17,18,19,20]. This underscores the imperative for further investigation and a deeper understanding of the pharmaceutical potential inherent in endophytic fungi [21].

Many endophytic microorganisms that peacefully coexist within plant tissues without harming their host plants [22] possess a remarkable ability to synthesize unique secondary metabolites [23]. Notably, fungi associated with marine macroalgae inhabit distinct ecological niches characterized by prolonged exposure to sunlight, fluctuating moisture levels, high salt concentrations, shifting tides, and the presence of insect herbivores [24,25,26]. To thrive under these challenging conditions, these fungi have developed mechanisms for producing novel secondary metabolites. These compounds hold significant promise for various applications owing to their specific adaptations to dynamic and extreme marine environments [27,28]. These fungi have gained recognition as prolific producers of a diverse range of bioactive compounds, including, but not limited to, anticancer agents, antibiotics, anti-angiogenic substances, antiviral compounds, and compounds with notable antiproliferative properties [29,30]. Of particular significance is the discovery of a significant number of antioxidant compounds originating from marine fungi. These antioxidants possess distinct properties that render them suitable for use across various sectors, including the food, cosmeceutical, and pharmaceutical industries [31]. Beyond their implications for human health, our research delves into the realm of aquaculture, where antimicrobials have traditionally played a critical role in combating pathogenic infections. However, the widespread application of antimicrobials in aquaculture carries extensive environmental consequences, affecting various bacterial species and facilitating the dissemination of genes associated with bacterial resistance [31,32,33,34]. In the pursuit of alternative approaches to combat pathogenic infections, our study highlights probiotics and immunostimulants as effective options in aquaculture settings. A significant diversity of epiphytic and endophytic fungi has been identified in association with the red algae *Pterocladiella capillacea*, encompassing 129 species from 67 genera. Hypocreales and Pleosporales dominated the community, alongside a notable presence of basidiomycetous yeasts (Sporidiobolales) and a smaller representation of Mucoromycota. Among the cultured fungi, we examined six fungal strains associated with red algae collected from Northern Taiwan [35] to assess their antioxidant activity potential and metabolic profiling. This approach seeks to enhance the growth and health of cultured organisms, which is particularly significant given the rise of antibiotic resistance stemming from the use of antibiotics and vaccines [34]. Essentially, our study aimed to advance sustainable aquaculture practices by exploring alternative antimicrobial solutions and tapping into the bioactive potential found within marine fungi [28,32,35].

## 2. Materials and Methods

### 2.1. Isolation of Marine Fungi

The species selected for this study, specifically NTOU4274, NTOU4203, NTOU4464, NTOU4446, NTOU4253, and NTOU 4416, were sourced from the Marine Fungi Laboratory at the Institute of Marine Biology, National Taiwan Ocean University. These specimens were primarily cultured from the red algae *Pterocladiella capillacea* at Chao-Jin Park. The sampling site, located at Chao-Jin Park (25°08′31.9″ N, 121°48′08.7″ E), is situated on the eastern side of the Badouzi Peninsula, affording a view of Wanghaixiang Bay in Keelung, northern Taiwan. The isolation and identification of these fungi associated with macroalgae have also been presented in previously published reports [35]. Sterilized/washed algal segments from both healthy and deceased thalli were cultured in glucose–yeast extract–peptone seawater agar, resulting in the isolation of pure fungal cultures. The identification process primarily relied on BLAST search analysis of the internal transcribed spacers of ribosomal DNA (ITS).

### 2.2. Preparation of Fungal Crude Extracts

The fungi were cultivated in 250 mL conical flasks containing GYSP medium, which consists of glucose, yeast, peptone, and sea salt. The cultivation was performed in a shaking incubator at 37 °C for 7 days [30]. After this incubation period, we visually examined the fungal colonies to validate culture purity. Subsequently, the cultures were aseptically transferred into 50 mL conical tubes. To initiate the extraction process, we centrifuged the fungal cultures at 5000 rpm for 10 min. The resulting filtrate was dissolved in 100 mL of cold methanol. This methanol–filtrate mixture was subjected to ultra-sonication for 30 min and then allowed to macerate for 1 day, with daily homogenization for 30 min using a magnetic stirrer, all performed at room temperature. After this maceration, the homogenate was dried using a rotary vacuum evaporator until the solvent was completely removed.

### 2.3. Determination of Antioxidant Activity

#### 2.3.1. Free Radical 1,1-Diphenyl-2-picryl-hydrazyl (DPPH)-Scavenging Test

The assessment of the overall antioxidant capacity of fungal extracts was conducted via the DPPH radical scavenging assay, following the methodology outlined by Paduch et al. [36] with minor modifications. This method relies on the ability of DPPH to undergo a color change in the presence of antioxidants. In brief, 100 μL of the test compound, ranging from 6.25 to 800 μg/mL (6.25, 12.5, 25, 50, 100, 200, 400, and 800 μg/mL), was mixed with 0.1 mL of freshly prepared 0.2 mM DPPH solution (0.2 mg/mL dissolved in methanol). Positive controls, such as ascorbic acid (vitamin C), known for their potent antioxidant properties, were employed. Ascorbic acid was used as the reference standard at concentrations of 6.25, 12.5, 25, 50, 100, 200, 400, and 800 μg/mL. A blank solution containing an equal amount of DPPH and methanol was prepared. The sample solutions were incubated in the dark for 30 min. The absorbance at 517 nm was then measured after incubation at room temperature. DPPH scavenging activity (%) was determined using the following formula:$$\mathrm{DPPH}\;\mathrm{scavenging}\;\mathrm{effect}\;(\%)=\frac{A_{control}-A_{sample}}{A_{control}}\;*\;100$$

A_control_ means the absorbance of the control sample, and A_sample_ means the absorbance of the standard or tested compound. The antioxidant capacity of the sample was expressed as IC_50_. This assay was performed in triplicate.

#### 2.3.2. 2,2′-Azinobis-(3-ethylbenzothiazoline-6-sulfonic acid) (ABTS) Radical-Scavenging Test

The antioxidant capacities of fungal extracts were determined using the ABTS method with a few modifications, following the protocol outlined by Sun et al. [37]. To generate ABTS+ ions, we added potassium persulfate (K_2_S_2_O_8_, Merck KGaA, Darmstadt, Germany) to ABTS, thoroughly mixed, and allowed to incubate in darkness at room temperature for 16 h. Stock solutions were prepared by dissolving 7.4 mM ABTS and 2.6 mM K_2_S_2_O_8_ in deionized water, combining them in a 1:1 ratio, and incubating the mixture in darkness for 16–24 h. Subsequently, 1 mL of the stock solution was diluted with 50 mL of deionized water until an absorbance of 1.1 ± 0.002 at 734 nm was achieved. Ascorbic acid (Sigma-Aldrich, St. Louis, MO, USA) was employed as the standard and mixed with 3 mL of the diluted ABTS+ solution. The scavenging activity of fungal extracts was evaluated by measuring the percentage of decolorization at 734 nm after 2 min of the reaction at room temperature. The ABTS+ scavenging activity (%) was calculated using the following formula: ((OD734control × OD734sample)/OD734control) × 100%. This assay was conducted in triplicate.

### 2.4. Determination of Total Phenolic Content

Total phenolic content in fungal extracts was determined utilizing the Folin–Ciocalteu method, as outlined by Orak [38], with minor modifications. A fungal homogenate (0.5 mL) was mixed with distilled water (8 mL), Folin–Ciocalteu reagent (0.5 mL, Merck KGaA), and sodium carbonate (Na_2_CO_3_, 1 mL of 7.5%, Merck KGaA). The resulting mixture was incubated at room temperature for 30 min, and its absorbance was subsequently measured at 765 nm using a spectrophotometer. A standard curve was constructed employing gallic acid as the reference. The entire assay was conducted in triplicate to ensure precision and reproducibility. This methodology conformed to established protocols for phenolic content assessment, thereby contributing to the scientific rigor of this research.

### 2.5. Determination of Total Flavonoid Content 

The flavonoid content in fungal extracts was quantified using a spectrophotometric method involving aluminum chloride (AlCl_3_), following the protocol established by Ghasemzadeh et al. [39]. In brief, fungal homogenate (1 mL) was blended with 5% sodium nitrite (NaNO_2_, 0.7 mL, Merck KGaA) and 30% ethanol (10 mL) for 5 min. Subsequently, 10% AlCl_3_ (0.7 mL, Merck KGaA) was added and thoroughly mixed. After 6 min, 1 mol/L sodium hydroxide (NaOH, 5 mL, Merck KGaA) was added, and the mixture was diluted to 25 mL with 30% ethanol and then left to stand for 10 min. The absorbance of the resulting solution was measured at 430 nm using a spectrophotometer (Shimadzu, Kyoto, Japan). To determine the flavonoid content accurately, we constructed a standard curve utilizing quercetin as the reference. This comprehensive assay was performed in triplicate.

### 2.6. Determination of Total Tannin Content

The tannins in fungal extracts were quantified using a colorimetric method involving the Folin–Denis reagent, as outlined by Chanwitheesuk et al. [40]. In brief, a fungal homogenate (0.5 mL) was mixed with 8 mL of distilled water, 0.5 mL of Folin–Denis reagent (Merck KGaA), and 1 mL of Na_2_CO_3_ (Merck KGaA). The resulting solution was thoroughly mixed and incubated at 27 °C for 30 min. The absorbance of the solution was subsequently measured at 760 nm, employing a tannic acid solution (Sigma-Aldrich) as the standard solution for calibration. This assay was meticulously performed in triplicate to ensure both the accuracy and reliability of the results, thus contributing to the scientific rigor of the study.

### 2.7. Antibacterial Activity

The antimicrobial efficacy of the fungal extract was evaluated against *Bacillus subtilis*, *Escherichia coli*, *Staphylococcus aureus*, *Enterobacter aeruginosa*, and *Vibrio parahaemolyticus*, all acquired from our laboratory. The assay involved impregnating 6 mm diameter filter paper disks with 100 µL of fungal filtrate extracts. These loaded disks were strategically placed on LB agar medium for antibacterial. Subsequently, the plates were incubated at 28 °C for 3 days to promote fungal growth and at 37 °C for 24 h to facilitate bacterial growth. After the respective incubation periods, we visually examined the zones of inhibition surrounding the disks and measured their diameters in millimeters. Notably, as controls, disks laden with ampicillin were employed as a reference antibacterial agent. This meticulous methodology provided valuable insights into the antimicrobial potential of the fungal extracts against a range of clinically relevant microorganisms.

### 2.8. Brine Shrimp Assay

The brine shrimp hatchability test utilized in this study was adapted from the method described by Lieberman et al. [41]. This investigation aimed to analyze the potential of marine secondary metabolites for aquaculture cost-effectively. The absence of a continuous cell line culture for shrimps hampers the timely development of immunostimulants and vaccines. To ensure that the observed mortality in the bioassay resulted from bioactive compounds rather than starvation, we compared the mortality in each treatment group. It is important to note that hatched brine shrimp *A. nauplii* can survive for up to 48 h without external food [42], as they continue to feed on their yolk sac during this period.

To assess the hatchability of brine shrimp cysts, we followed the subsequent procedure, albeit with a few modifications to enhance standardization. Initially, 0.5 g of dried brine shrimp cysts were collected and weighed. These prepared cysts were immersed in seawater at a concentration of 1 g of cysts per liter. The hatching process was conducted at 28 °C, with continuous illumination and robust aeration to ensure optimal hatching conditions. For each experimental trial, groups of 10 *Artemia nauplii*, aged 12 h, were individually transferred to designated wells using appropriate pipettes. Subsequently, *A. nauplii* was exposed to varying concentrations of the fungal extracts being tested.

The lethality test for fungal extracts employed brine shrimp eggs, which were hatched in artificial seawater and utilized 48 h post-hatching. The assay adhered to the protocol outlined by Meyer et al. [42]. Various sample concentrations (1,10, 100, and 1000 µg/mL) were prepared in triplicate, transferred to glass vials, and subsequently evaporated. Artificial seawater was then added to each vial to achieve the desired concentration. Thirty brine shrimp were introduced into each vial, and the number of deceased shrimps per dosage was recorded after 24 h. The survival rate of the shrimp in all vials, including the positive control (etoposide), was meticulously documented. This method facilitated the assessment of the potential cytotoxicity of the fungal extracts, providing valuable insights into their impact on brine shrimp survival as a preliminary indicator of potential biological activity.

Furthermore, the percentage of hatch inhibition (%HI) was calculated by subtracting the hatchability percentage of each treatment from that of the control group, as indicated by the following formula: %HI = %hatchability in the control − %hatchability in each treatment.

### 2.9. LC-HDMSE Analysis of Fungal Extracts

High-resolution and high-mass-accuracy LC-MS analyses were performed using an ACUITY UPLC I-Class system (Waters, Milford, MA, USA) coupled to a SELECT SERIES Cyclic IMS instrument (Waters, Wilmslow, UK), featuring a Q-ToF mass spectrometer equipped with a cIM device. Chromatographic separation was performed using an ACQUITY UPLC CSH C18 peptide column (1 mm I.D. × 150 mm, 5 µm particle size, 100 Å pore size) maintained at 45 °C. Mobile phase A consisted of water with 0.1% formic acid (*v*/*v*), while mobile phase B consisted of acetonitrile with 0.1% formic acid (*v*/*v*). The LC system was operated at a flow rate of 50 uL/min, utilizing a linear gradient elution program that was initiated with 2% buffer B at 2 min and reached 98% buffer B at 50 min. The HDMSE instrument was equipped with an electrospray ion source, operating at a capillary voltage of 2.5 kV in positive ion mode. The desolvation gas flow was set to 800 L/h, with the desolvation gas temperature maintained at 300 °C, and the source temperature was 100 °C. Data were acquired in the m/z range of 50 to 1200 with a 1 s scan time. The mass instrument was operated in the LC-HDMSE mode of data acquisition, employing alternating 1 s scans for low (4 V) and high (10–40 V) collision energies while incorporating lock mass and a separation time of 5 ms. The LC-MS instrument was controlled using MassLynx software (ver. 4.2). 

#### Metabolites Analysis

Progenesis QI (Nonlinear Dynamics, Durham, NC, USA) was employed to process the raw MS data. Spectral matching was performed using Progenesis MetaScope (version 1.0.6901.37313), employing a metabolite structure database (SDF format) obtained from the Human Metabolome Database (HMDB, version 5). The search parameters comprised a precursor mass tolerance of 12 ppm, a theoretical fragmentation mass tolerance of 12 ppm for fragment searching, and a compound identification threshold value of 30. The abundance of identified metabolites was determined by Progenesis QI. 

### 2.10. Statistical Analysis

Each and every result was given as mean ± standard deviation (SD). When feasible, an ANOVA followed by a Tukey post hoc test was used to analyze all of the data derived from chemical quantifications. A significance threshold of *p* < 0.05 was applied. SPSS Statistics 24 (SPSS, Chicago, IL, USA) was used for all statistical analyses.

## 3. Results

### 3.1. Antioxidant and Phytochemical Analysis 

The antioxidant activity and phytochemical content analyses of fungal extracts revealed notable variations among the studied species. The evaluation of ABTS+ and DPPH-scavenging activities, as well as the quantification of total phenolics, flavonoids, and tannins, provided comprehensive insights into the antioxidant potential of fungal extracts, with ascorbic acid serving as a reference. Concentrations of ascorbic acid ranged from 6.25 to 800 μg/mL. At a concentration of 100 μg/mL, ascorbic acid demonstrated an IC_50_ value of 7.03 ± 2.8. Notably, as shown in Table 1, *Trichoderma harzianum* exhibited the lowest IC50 value in the ABTS+ assay, indicating the highest antioxidant activity (25.01 ± 3.18). Similarly, in the DPPH assay, *T. harzianum* displayed the lowest IC50 value (12.2 ± 2.6). In contrast, both *Hypomontagnella monticulosa* and *Hortaea werneckii* exhibited relatively higher IC_50_ values in both assays, indicating comparatively lower antioxidant activities. Concerning total phenolics, *T. harzianum* exhibited the highest content (93.01 ± 1.8 mg/100 g), while *H. monticulosa* had the lowest (23.004 ± 2.8 mg/100 g). The highest flavonoid content was observed in *Aspergillus* sp.7 (43.06 ± 0.02 mg/100 g), while *H. monticulosa* showed the lowest (18.24 ± 0.2 mg/100 g). Notably, *T. harzianum* possessed the highest total tannin content (9.002 ± 2.5 mg/100 g), while *H. monticulosa* displayed the lowest (16.03 ± 4.1 mg/100 g). These results highlight the diverse antioxidant profiles among the fungal extracts, with *T. harzianum* emerging as a particularly promising source of potent antioxidants.

### 3.2. Brine Shrimp Lethal Assay

The crude extracts of endophytic marine fungi, assessed at various concentrations (1, 10, 100, and 1000 µg/mL), demonstrated minimal toxicity towards brine shrimps (Figure 1) ranging from 1 µg/mL to 1000 µg/mL exhibited an average mortality rate of 26.75%, while *T. harzianum* exhibited approximately 30.2% mortality. Etoposide was used as the standard drug for the cytotoxicity test. The survival rate of the control etoposide was reported as 86%. 

### 3.3. Antibacterial Activity

The antibacterial activity of fungal extracts against various bacterial strains was evaluated, and the results are presented in Table 2. Notably, *H. werneckii* demonstrated significant inhibitory effects against *B. subtilis*, *V. parahaemolyticus*, and *E. coli*, with inhibition values of 1.4, 1.2, and 0.1, respectively. *T. harzianum* also exhibited notable antibacterial activity, particularly against *V. parahaemolyticus* and *E. coli*, displaying inhibition values of 2.4 and 1.2, respectively. *E. maritima* displayed moderate inhibitory effects against *S. aureus* (3.1) and *E. coli* (0.2). In contrast, *Engyodontium album* exhibited substantial inhibition against *B. subtilis* (1.2) and *S. aureus* (1.0), *Bacillus subtilis*, *Escherichia coli*, *Staphylococcus aureus*, *Enterobacter aeruginosa*, and *Vibrio parahaemolyticus*.

### 3.4. LC-HDMSE

Niazicin A, sucrose octaacetate, hemorphin-4, and approximately 8000 other compounds were identified via LC-HDMSE spectra analysis (Table 3) utilizing the HMDB. Appendix A contains the LC-MS analysis and major compounds present in the fungal extracts. While Appendix A is an extended version of Table 3, it shows the specific functional group information about the metabolites, complementing the overall idea conveyed by Appendix A. Notably, the primary secondary metabolites found in all six fungi exhibiting the highest antioxidant activity, as detailed in Table 2, predominantly fell within the superclass of organic oxygen compounds (Figure 2a). Subsequent tentative identification of these crucial features using LC-HDMSE in conjunction with the HMDB database revealed their classification into the following categories: organic oxygen compounds, lipids, organic acids and their derivatives, organoheterocyclic compounds, phenylpropanoids and polyketides, benzenoids, lignans, neolignans, and related compounds, alkaloids and their derivatives, organic nitrogen compounds, as well as nucleosides, nucleotides, and their analogues (Figure 2a). Venn diagrams (Figure 2b) show the number of compounds identified in the six fungal extracts.

Various classes of metabolites were identified in these species, including organooxygen compounds, carboxylic acids and their derivatives, steroids and steroid derivatives, benzene and its substituted derivatives, prenol lipids, benzoxazines, fatty acyls, diazinanes, lipids and lipid-like molecules, lignan glycosides, pyridopyrimidines, flavonoids, cinnamic acids and their derivatives, organonitrogen compounds, naphthalenes, peptidomimetics, cytochalasans, glycerophospholipids, quinolines and their derivatives, azoles, yohimbine alkaloids, indoles and their derivatives, phenanthrenes and their derivatives, macrolides and analogues, stilbenes, macrolactams, piperidines, organic phosphonic acids and their derivatives, rofurans, pyrrolidines, phthalide isoquinolines, dibenzylbutane lignans, camptothecins, triazines, anthracenes, sphingolipids, isoquinolines and their derivatives, biotin and its derivatives, keto acids and their derivatives, morphinans, and tetracyclines. Appendix A is an extended version of Table 3; it shows the specific functional group information about the metabolites, complementing the overall idea conveyed by Appendix A. The functional group information of specific metabolites can be easily identified from the figures provided.

## 4. Discussion

In this study, we evaluated the antioxidant capacity and antioxidant composition of six fungi associated with the red algae *P. capillacea*, collected from northern Taiwan. The findings revealed that *E. maritima*, *E. album*, *H. monticulosa*, *H. werneckii*, *T. harzianum*, and *Aspergillus* species exhibited antioxidant potential, effectively scavenging ABTS+. Additionally, the fungal isolates demonstrated robust antioxidant capabilities, as evidenced by IC50 values less than 100 μg/mL in the DPPH assay. Remarkably, the findings underscored the significantly superior antioxidant capacity of *T. harzianum* compared to *E. maritime*, *E. album*, *H. monticulosa*, *H. werneckii*, and *A. niger*. The increasing emphasis on microorganisms holds great promise for addressing the demands of industries in search of novel and abundant sources of bioactive compounds. The findings of our study reveal noteworthy variations in the antioxidant activities and phytochemical profiles of six fungal species isolated from marine environments [43,44,45,46,47]. 

*T. harzianum* stood out as a prominent candidate, demonstrating exceptional antioxidant potential with the lowest IC_50_ values in both ABTS+ and DPPH assays. This suggests its suitability as a viable source of potent natural antioxidants. *E. maritime* and *E. album* also exhibited significant antioxidant activity, highlighting their bioactive potential. Furthermore, *T. harzianum* showcased the highest total phenolic content, emphasizing its abundance of health-promoting phenolic compounds. These findings underscore the remarkable diversity of bioactive compounds in marine-derived fungi, with implications for their utilization in pharmaceuticals, the food industry, and natural product research [48]. Our findings emphasize the pivotal role played by fungal metabolites in advancing the development of safer and more efficacious food additives and pharmaceutical ingredients [4,49,50,51,52,53]. This aligns with the increasing interest in harnessing natural products for various applications. Our study also revealed that most of these fungi exhibit notable antioxidant potential, as evidenced by their effective scavenging of ABTS+ radicals. Taken together, our findings indicate that, in addition to their ability to release antioxidant compounds from plant cell walls [18,19,20,21]. endophytic marine fungi may inherently serve as valuable reservoirs of substances associated with antioxidant activity [54]. Numerous studies have indicated that the antioxidant capacity of filamentous fungi is primarily associated with their phenolic content [18,19,55]. It tends to increase with a higher total phenolic content. Flavonoids, the most extensive group of phenolic compounds, are widely prevalent in human diets. Oranges emerge as a substantial source of flavonoids, containing approximately 69.85 mg/100 g. Notably, the flavonoid content of the fungal extracts examined in this study was comparable to that found in oranges. The association between the presence of flavonoids and the antioxidant potential of food items has been well recognized [56,57,58,59,60,61,62,63,64,65]. The exceptional antioxidant capability of *T. harzianum* places it in league with a well-established benchmark antioxidant: ascorbic acid. Ascorbic acid is widely recognized for its potent natural antioxidant properties and serves as a prominent representative of dietary antioxidants for human health in ABTS assay and DPPH assays, thus emphasizing the remarkable performance of *T. harzianum* in this regard [66,67,68,69]. Notably, *Aspergillus* sp.7 did not exhibit measurable inhibitory activity against the tested bacterial strains. These results highlight species-specific variations in the antimicrobial potential of the fungal extracts, suggesting their potential application in the development of novel antimicrobial agents.

Various classes of metabolites were identified in these fungal extracts (Figure 2). While the classification of abundant metabolites are at the superclass level, the major groups lie under the category of alkaloids and derivatives, benzenoids, lignans, neolignans and related compounds, lipids and lipid-like molecules, nucleosides, nucleotides and analogues, organic acids and derivatives, organic nitrogen compounds, organic oxygen compounds, organo-heterocyclic compounds, phenylpropanoids, and polyketides. The Venn diagram illustrates the distribution of metabolites identified via LC-HDMSE analysis. Each circle denotes the total count of metabolites detected within the respective fungal extracts, while the overlapping areas signify the shared metabolites across the analytical methods. The LC-HDMSE analysis identified approximately 8000 metabolites in total, with 1280 metabolites utilized for classification based on their abundance. Numerous compounds were isolated from various fungal strains. Among them, niazicin A stood out as one of the most abundant compounds in *E. maritima*. Moreover, niazicin A, as reported in the literature, exhibits anti-inflammatory activity, as demonstrated in mouse RAW264.7 cells [56], where its impact was assessed by inhibiting LPS-induced nitric oxide production. The evaluation specifically involved a 30 min incubation period preceding an LPS challenge. The documented anti-inflammatory properties of niazicin A significantly enhance its relevance within FooDB, an open-access database that contains chemical composition data for typical, unprocessed foods. This underscores the database’s commitment to encompass not only the structural and compositional aspects of compounds but also their functional and physiological attributes. This information contributes to a broader understanding of the potential health effects of compounds found in natural products such as *Moringa oleifera* [70].

The identification of these bioactive compounds within fungal extracts, characterized by their remarkable antioxidant and antibacterial properties, represents a significant advancement in the field of natural product exploration [54,71]. Foremost among these discoveries was the isolation of niazicin A from fungal sources, which holds immense potential in drug development. Additionally, the notable antibacterial attributes of niazicin A have substantial implications in addressing the prevailing challenge of antibiotic resistance [67]. This underscores the potential of niazicin A to serve as a valuable innovative antibiotic. Investigating this compound offers a promising avenue for developing novel therapeutic agents to address pressing global health concerns. Another noteworthy discovery within the fungal extracts was sucrose octaacetate, which was present in all six samples and exhibited antioxidant properties. This compound has versatile applications in both the food and pharmaceutical industries. As a sucrose derivative, its significance extends to its potential as a sweetening agent and an excipient in pharmaceutical formulations [58,59,60,61,62]. The revelation of this compound highlights the pivotal role of natural products in advancing the development of safer and more efficacious food additives and pharmaceutical ingredients, further underscoring the potential of these fungal-derived compounds. The discovery of this compound emphasizes the crucial contribution of natural products to the progress of creating safer and more effective food additives and pharmaceutical components, reaffirming the promise held by these compounds derived from fungi [55,72]. An exploration of sucrose octaacetate’s safety profile and functional properties holds the potential to play a significant role in the development of healthier food and pharmaceutical products. Hemorphin-4, a unique peptide derived from hemoglobin, presents an intriguing enigma concerning its physiological function [58]. Hemorphins, including hemorphin-4, comprise a class of peptides with elusive roles within the body. Investigating hemorphin-4’s role can yield valuable insights into complex physiological processes, particularly those associated with the blood and neural systems. Such comprehension may pave the way for therapeutic interventions in disorders associated with these intricate systems, making its discovery of paramount importance in the fields of neuroscience and physiological research. Hemorphins are peptides with potential implications for both the blood and neural systems, and the study of hemorphin-4 may provide insights into their physiological functions. Understanding the role of hemorphin-4 can contribute to unraveling complex physiological processes and may have implications for therapeutic interventions in disorders related to the blood and neural systems. 

In chromatography and biotechnology, phenyl agarose plays a pivotal role in protein purification [59,60,61]. Its importance is underscored by its use as a matrix in chromatographic separations to isolate proteins. Its significance in biotechnology and biochemistry research cannot be overstated, as it is a fundamental tool for purifying bioactive compounds. Its role in facilitating the production of pure proteins for various applications is crucial for advancing biotechnological and biomedical research. The compound (3b,20R,22R)-3,20,27-trihydroxy-1-oxowitha-5,24-dienolide 3-glucoside exhibits potential pharmacological relevance [73]. Its isolation from fungal sources implies potential traditional uses in herbal medicine. Investigating its bioactivity and understanding its role in traditional healing practices are essential for preserving and effectively utilizing this natural resource. The study of this compound aligns with the growing interest in ethnobotany and the development of plant-based therapeutics. In the field of cardiology, the discovery of digitoxigenin bisdigitoxide and digitoxigenin 3-[glucosyl-(1->6)-glucosyl-(1->4)-2,6-dideoxyribohexoside] is noteworthy. These derivatives of digitoxigenin, a well-known cardiac glycoside, hold significance [56,74,75,76]. Cardiac glycosides boast a rich historical legacy in the treatment of heart-related conditions. The identification and examination of these derivatives may yield insights into their cardiac activity, thereby potentially presenting alternative or complementary therapeutic avenues for heart-related disorders. In the realm of lipidomics and cell membrane investigation, lysoPS (16:0/0:0) and lysoPG (18:1(9Z)/0:0) hold significant prominence as constituents of cellular membranes [61]. Their roles in governing membrane composition and fluidity play a central role in cellular physiology. Thorough research into these compounds is imperative for a deeper understanding of cell membrane function and its implications for various pathological conditions, including cancer and neurodegenerative diseases [72,77,78]. 

The IUPAC (International Union of Pure and Applied Chemistry) name for Chembl4211493 is *N*-(4,6-dimethylpyrimidin-2-yl)-4-[2-(4-methoxy-3-methylphenyl)-5-(4-methylpiperazin-1-yl)-4,5,6,7-tetrahydroindol-1-yl] benzenesulfonamide. It is known as phosphodiesterase 7A, an enzyme that plays a crucial role in numerous vital physiological and pathological processes by modulating intracellular cyclic adenosine monophosphate signaling [79]. Research has provided compelling evidence of PDE7’s widespread expression in the central nervous system (CNS), suggesting its intricate involvement in various CNS functions and potentially influencing the pathogenesis of several neurological diseases [80]. Sucrose octaacetate serves as a versatile component in pesticide products, functioning as an inert ingredient in insect repellents, herbicides, flea and tick sprays, and various other insecticides [58]. Sucrose octaacetate, aside from its role in pest control, finds diverse commercial applications, including impregnating and insulating papers, as well as contributing to the formulation of lacquers and plastics [75]. It possesses a molecular formula of C_28_H_38_O_19_ and a molecular weight of 678.6 g/mol, with its IUPAC name being [(2R,3R,4S,5R,6R)-3,4,5-triacetyloxy-6-[(2S,3S,4R,5R)-3,4-diacetyloxy-2,5-bis(acetyloxymethyl)oxolan-2-yl] oxyoxan-2-yl] methyl acetate. Sucrose octaacetate is recognized for its hygroscopic solid physical state. The extensive regulatory approval granted to sucrose octaacetate by the US Food and Drug Administration (FEMA No. 3038) [76] further highlights its significance. It has received approval for use both as a direct and indirect food additive, reaffirming its safety for applications in the food industry. Additionally, sucrose octaacetate has been authorized for use in over-the-counter drug products as a deterrent for nail biting and thumb sucking, demonstrating its utility beyond agriculture. This multifaceted approval highlights the compound’s versatility and safety across various domains, making it a valuable and sanctioned ingredient in various consumer and industrial products.

Hemorphin-4, an endogenous opioid peptide composed of 4 amino acid residues, belongs to the hemorphin family and possesses antinociceptive properties. It is derived from the β-chain of hemoglobin present in the bloodstream and exhibits the amino acid sequence Tyr-Pro-Trp-Thr [81]. Research indicates that hemorphin-4 displays affinities for μ-, δ-, and κ-opioid receptors, akin to structurally related β-casomorphins. However, it notably displays a higher affinity for the κ-opioid receptor. In addition to its role as a receptor agonist in the opioid system, it also exerts inhibitory effects on the angiotensin-converting enzyme [48]. The compound’s intricate molecular structure and multifaceted pharmacological profile contribute significantly to its importance in the field of opioid peptides and its potential therapeutic applications [61].

Comprising a phenyl group integrated with agarose, phenyl agarose serves as a versatile tool in various scientific applications. While it may not directly exhibit biological activity, its significance lies in facilitating the isolation of biologically active substances, including proteins and enzymes, enabling researchers to explore their structural and functional aspects. The compound’s chemical properties are characterized by a molecular formula of C_30_H_42_O_19_, a molecular weight of 706.6 g/mol, and its hydrophobic interactions originating from the phenyl group, rendering it effective for hydrophobic interaction chromatography. The compound (3b,20R,22R)-3,20,27-trihydroxy-1-oxowitha-5,24-dienolide 3-glucoside is classified as both a withanolide and a glycoside, with a molecular formula of C_34_H_50_O_11_ and a molecular weight of 634.8 g/mol [81].

Digitoxin digitoxoside, with a chemical formula of C_35_H_54_O_10_ and a molecular weight of 634.8 g/mol, belongs to the class of cardiac glycosides. This secondary metabolite is derived from *Digitalis lanata* and is also recognized by synonyms such as digitoxigenin bisdigitoxide. Biologically, digitoxin digitoxoside exhibits significant pharmacological effects, primarily within the cardiovascular system [48]. Comprehending its chemical properties, classification, and biological activities is crucial for its safe and efficient application in clinical settings.

PS (16:0/0:0), a 1-acyl-sn-glycero-3-phosphoserine, possesses a molecular formula of C_22_H_44_NO_9_P and a molecular weight of 497.6 g/mol. It is also known by the synonyms CHEMBL3577145 and LPS (16:0), indicating that it is a metabolite in *Saccharomyces cerevisiae* [66,82]. This compound possesses a complex structure composed of 33 heavy atoms and 24 rotatable bonds. It exhibits a calculated XLogP3-AA value of 2 and has a topological polar surface area of 166 Å^2^. Beyond its structural characteristics, PS (16:0/0:0) has been examined via various bioassays, displaying agonist activity at P2Y10, GPR34, and GPR174 receptors, thus shedding light on its potential pharmacological significance [62].

The pursuit of identifying novel therapeutic compounds has been a driving force behind extensive research efforts throughout human history. Given that more than 70% of the Earth’s surface is covered by water, marine ecosystems provide a vast array of unique and diverse environments. These ecosystems harbor a remarkable variety of organisms, many of which remain unexplored. Thanks to recent technological advancements, we now possess the means to discover and harness the potential of these organisms as valuable sources of a wide range of natural products [18].

## 5. Conclusions

In this investigation, we successfully isolated six marine fungi associated with red seaweed, all of which demonstrated remarkable antioxidant activity. While extensive research efforts are currently focused on harnessing bioactive compounds for aquaculture, limited exploration has been conducted regarding marine fungi as potential sources of immunostimulants for shrimp. Our findings suggest that these less common fungal varieties and species may possess even greater potential in terms of antioxidant and antibacterial properties compared to their conventional counterparts. Given the absence of a specific shrimp cell line, we employed brine shrimp for preliminary cytotoxicity assessments prior to conducting animal experiments. Additionally, these fungi exhibited significant antibacterial activity against aquaculture pathogens and were found to be rich in phenolic, tannin, and flavonoid content. This discovery not only enhances our understanding of natural compounds but also presents promising prospects for their application in the field of aquaculture as immunostimulants. Our study lays the groundwork for future investigations into the potential applications and underlying mechanisms of these compounds, which hold the potential to advance both human health and scientific understanding.

By gaining a deeper understanding of these fungal communities and their applications, we aimed to contribute to the development of effective and environmentally responsible approaches that enhance shrimp health and productivity. Our findings potentially lay the groundwork for identifying suitable candidates for the development of fungal-based immunostimulants for shrimp.

## Figures and Tables

**Figure 1 antioxidants-13-00336-f001:**
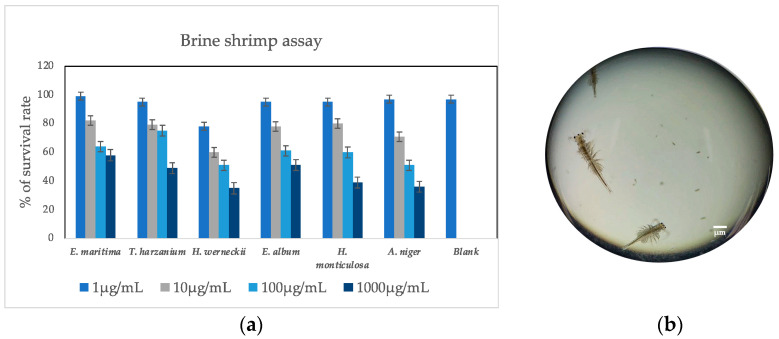
(**a**) The cytotoxic effect of crude fungal extracts in comparison with blank (no extract added) on brine shrimp for 48 h. (**b**) Brine shrimp under a microscope.

**Figure 2 antioxidants-13-00336-f002:**
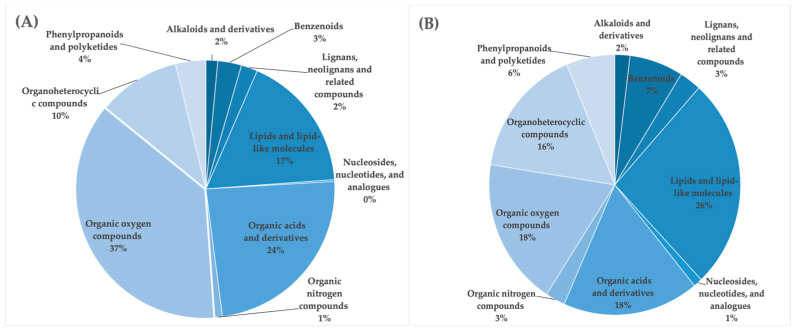
Superclass-level classification of fungal metabolites obtained from LC-HDMSE. (**a**) Superclass-level classification of major metabolites obtained from fungal metabolites. (**A**) *E. maritima*, (**B**) *E. album*, (**C**) *H. monticulosa*, (**D**) *H. werneckii*, (**E**) *T. harzianum*, and (**F**) *Aspergillus* sp.7. (**b**) Venn diagrams showing the number of compounds identified in the six fungal extracts. The overlapping area of the Venn diagram denotes the shared compounds. Each color indicates a different species: A (blue) *E. maritima*, B (red) *E. album*, C (green) *H. monticulosa*, D (yellow) *H. werneckii*, E (brown) *T. harzianum*, and F (turquoise) *Aspergillus* sp.7.

**Table 1 antioxidants-13-00336-t001:** IC_50_ Values of the antioxidant activities (DPPH and ABTS) and total phenolic, flavonoid, and tannin content of six marine fungi associated with the red algae *P. capillacea*.

Samples	ABTS+ Scavenging Activity (IC_50_ µg/mL)	DPPH Scavenging Activity (IC_50_ µg/mL)	Total Phenolic Content (mg/100 g)	Total Flavonoid Content (mg/100 g)	Total Tannin Content (mg/100 g)
*Emericellopsis maritima*	38.01 ± 0.2	16.5 ± 1.2	85.007 ± 2.6	26. ± 04	8.03 ± 1.4
*Engyodontium album*	52.00 ± 1.4	26.8 ± 0.24	64.02 ± 1.4	32 ± 001	4.002 ± 2.01
*Hypomontagnella monticulosa*	78.64 ± 2.6	34.7 ± 2.1	23.004 ± 2.8	18.24 ± 02	16.03 ± 4.1
*Hortaea werneckii*	81.02 ± 1.12	35.6 ± 4.8	48.03 ± 0.2	34.01 ± 01	7.031 ± 0.026
*Trichoderma harzianum*	25.01 ± 3.18	12.2 ± 2.6	93.01 ± 1.8	31.32 ± 02	9.002 ± 2.5
*Aspergillus* sp.7	76.24 ± 4.6	41.01 ± 1.4	65.02 ± 2.6	43.06 ± 002	24.1 ± 4.2
Ascorbic acid	7.03 ± 2.8				

The results are expressed as the mean ± standard deviation of data from three experiments (*n* = 3). No significant changes were noted.

**Table 2 antioxidants-13-00336-t002:** Antibacterial activity (zone of inhibition) recorded in centimeters.

Species	*Bacillus subtilis*	*Vibrio parahaemolyticus*	*Escherichia coli*	*Staphylococcus aureus*	*Enterobacter aeruginosa*
*Emericellopsis maritima*	0.01	1.4	0.2	3.1	0.2
*Engyodontium album*	1.2	0.6	0.8	1.0	0
*Hypomontagnella monticulosa*	0	0.2	0	1.2	0
*Hortaea werneckii*	1.4	1.2	0.1	0	0
*Trichoderma harzianum*	1.2	2.4	1.2	0.4	0.01
*Aspergillus* sp.7	0.2	0	0	0	0

**Table 3 antioxidants-13-00336-t003:** Abundance of metabolites in the fungal extracts.

Compounds	*E. maritima*	*E. album*	*H. monticulosa*	*H. werneckii*	*T. harzianum*	*A. niger*	Compound ID
Niazicin A	155,202.77	673.01	7161.57	2464.28	124,721	69.81158	HMDB0303690
Chembl4211493	97,804.71	173.16	120.93	46.04	119.58	52.16	HMDB0257612
Sucrose octaacetate	73,594.50	27,778.63	38,632.07	109,316.67	71,510.63	30,874.62	HMDB0029893
Hemorphin-4	62,948.70	233.88	205.95	68.27	201.67	144.13	HMDB0059788
Phenyl-agarose	16,089.59	2968.72	3263.43	7227.09	8201.72	3438.86	HMDB0256427
(3b,20R,22R)-3,20,27-Trihydroxy-1-oxowitha-5,24-dienolide 3-glucoside	12,936.76	496.91	241.33	601.10	85.78	310.62	HMDB0033573
Digitoxigenin bisdigitoxide	12,787.91	426.99	638.27	1280.91	3525.74	845.21	HMDB0251275
LysoPS (16:0/0:0)	11,497.57	219.47	3363.01	221.33	474.59	890.39	HMDB0240605
Sinalbine	11,455.53	247.90	1259.21	20,336.59	781.93	155.90	HMDB0303664
Olodaterol	7307.81	3584.23	2275.44	14,503.53	4670.76	4774.74	HMDB0255957
Motolimod	6348.01	423.97	1483.55	2991.55	1955.24	1470.58	HMDB0254904
Docosanamide	5797.60	576.58	661.097	1660.71	637.03	551.65	HMDB0000583
N-Nitrosofenfluramine	5669.05	3123.30	6295.30	28,770.25	5804.03	7980.27	HMDB0255206
Peonidin acetyl 3,5-diglucoside	5362.16	2356.22	2802.88	8467.53	4177.09	2270.36	HMDB0301895
Ophiopogonin C’	5341.08	429.59	191.01	518.264	2366.44	309.52	HMDB0029312
Aminopentol	5167.53	4490.50	14,032.33	17,448.69	2373.39	10,994.10	HMDB0248328
Ribociclib	4802.41	3.59	6.48	4.73	11.04	21.46	HMDB0257211
Suvorexant	4624.75	1166.79	236.07	4488.59	811.25	364.64	HMDB0258640
Janthitrem G	4280.60	5900.46	1006.90	11,179.17	1748.32	1830.12	HMDB0030531
Labadoside	3695.32	37.06	126.02	65.39318	4414.33	331.43	HMDB0036397
1-Sinapoyl-2,2′-diferuloylgentiobiose	3672.46	7.76	30.44	43.49	1617.97	1.75	HMDB0301721
Digitoxigenin 3-[glucosyl-(1->6)-glucosyl-(1->4)-2,6-dideoxyribohexoside]	3437.48	2702.59	2050.30	11,713.11	4860.62	1283.45	HMDB0034321
Boviquinone 4	3387.657	57.35	65.96	153.14	74.39	89.42	HMDB0030057

## Data Availability

The original data presented in the study are included in the article; further inquiries can be directed to the corresponding author.

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
