# Peer review of "A Comparison of the Antioxidant Potential and Metabolite Analysis of Marine Fungi Associated with the Red Algae Pterocladiella capillacea from Northern Taiwan"

_antioxidants, 2024, doi:10.3390/antiox13030336_

Round 1

Reviewer 1 Report

The manuscript of Jiji Kannan reports an investigation centered on screening marine fungi associated with the red algae Pterocladiella capillacea, which was collected from Chao-Jin Park in Keelung, Taiwan, as potential immunostimulants for shrimp aquaculture. In particular, the results regard antioxidant potential, brine shrimp lethal assay, antibacterial activity and LC-HDMSE analysis of fungal extracts.

   The manuscript has interesting data for publications. However, a number of doubts need to be clarified and listed below:

- In the image of figure 2b the size scale must be added

- The reference to tables 3 and 4 is missing in the main text

- The main text of paragraphs 3.5 and 3.6 is not collapsible. The text must be rewritten entirely. Have any sentences been lost in the formatting?

- The authors should provide more information on how they obtained the information on the abundances of the metabolites reported in Table 4

-I don't understand figure 4. The legend of figure 4 goes at the head of it.

Author Response

Reviewer 1

Comment 1: In the image of figure 2b, the size scale must be added

Response 1: Thank you for the reviewer's comment. Figure 2b was replaced by adding the scale bar.

Comment 2: The reference to tables 3 and 4 is missing in the main text

Response 2: We thank you for the reviewer's suggestion. New references were added according to reviewers’ suggestion.

Comment 3 The main text of paragraphs 3.5 and 3.6 is not collapsible. The text must be rewritten entirely. Have any sentences been lost in the formatting?

Response 4: We thank the reviewer for the relevant comment. The manuscript has been modified according to reviewer’s suggestions.

Comment 4 The authors should provide more information on how they obtained the information on the abundances of the metabolites reported in Table 4

Response 4: Thank you for the reviewer’s comment. We have modified the materials and methods section 2.9.1 Metabolites analysis. The updated section as follows:

2.9.1. Metabolites analysis

Progenesis QI (Nonlinear Dynamics, Durham, NC) was employed to process the raw MS data. Spectral matching was performed using Progenesis MetaScope (version 1.0.6901.37313), employing a metabolite structure database (SDF format) obtained from the Human Metabolome Database (HMDB, version 5). The search parameters comprised a precursor mass tolerance of 12 ppm, a theoretical fragmentation mass tolerance of 12 ppm for fragment searching, and a compound identification threshold value of 30. The abundance of identified metabolites was determined by Progenesis QI.

Comment 5: I don't understand figure 4. The legend of figure 4 goes at the head of it.

Response 5: Thank you for the reviewer's comment. While Figure 4 was an extended version of Table 4, it shows the specific functional group information about the metabolites, complementing the overall idea conveyed by Table 4.  The functional group information of specific metabolites can be easily identified from the figures provided. The manuscript has been modified by moving the Figure 4 to the supplementary information.

Reviewer 2 Report

This paper presents what could be interesting results, but those are very difficult to tease out relevant results from everything else.

Abstract does not present any results.

Line 44 - change 'mushrooms' to a specific class of fungi and provide a reference for the statement.

Line 66 - change to 'Many endophytic microorganisms that peacefully coexist...'

Lines 80-92 - delete - not relevant to the paper or move to conclusions

Lines 106-110 - move to conclusions

Line 113 - Need to provide identity of the fungi here. How were they cultured? How were they identified? How were they shown to be endophytes? Why were these fungi selected for the work presented here?

Line 127 - change to '...GYSP medium, which consists of ...'

Line 164 - give concentration of sodium carbonate used

Line 192-194 - italicize the bacterial names

Line 196-198 - Why was fungal testing included in the antibacterial section? No fungi were listed for testing.

Line 211-213 - italicize the scientific name for brine shrimp

Line 222 - need to explain the positive control

Line 261 - section 3.1 - delete section/photos - no morphological data was presented

Section 3.2 - change title to 'Antioxidant and Phytochemical Analyses'

 - need to link fungal names with identifiers used in materials section

Line 273 - need to write out Trichoderma

Line 276 - delete sentence '..reaffirming....capability.' This is a conclusion.

Table 1 - write out genus name and add identifiers

Line 289 - change to 'No significant differences were noted.' or add letters to the table

Line 279 - not what the table shows

Line 281 - Table 1 don't match - Aspergillus sp. 7 -vs- Aspergillus niger - change one of these

Line 294-301 - move to methods section

Figure 2 - Is the survival rate compared to  no extract added? If yes, write that in the legend.

Line 315-317 - delete or move to conclusions

Table 2 - Write out all genus names

Table 3 - move to supplemental  - need to explain column headings in legend

Delete figure 4 or move to supplemental

Lines 340-351 - Only include three important types of compounds.

Lines 356-366 - move to discussion

Lines 381-440 - move to discussion

Author Response

Reviewer 2

Comment 1: Does the title describe the article's topic with sufficient precision?

The authors never discussed any type of association with red algae.

Response 1: Thank you for the reviewer’s comment. We have revised the manuscript by adding the missing details.

Comment 2: Does the introduction provide a comprehensive yet concise overview about the state of knowledge in the area of research? Much of the introduction was a justification of the area of study and was not relevant to this paper.

Response 2: Thank you for the reviewer’s comment. we have been revised the manuscript by rewritten the introduction part by adding the relevant details citing new references in this field of study.

Comment 3: Is the research design appropriate and are the methods adequately described? Almost nothing is written about the fungi that were used, how they were identified etc.

Response 3: Thank you for the reviewer’s comment. we have been revised the manuscript by adding the missing data by the reviewer’s comment.

Comment 4: Are the results presented clearly and in sufficient detail, are the conclusions supported by the results and are they put into context within the existing literature?

Response 4: Thank you for the reviewer’s comment. We have revised the manuscript by citing relevant references in this field of study. We have modified the discussion and conclusion section according’s to reviewers’ suggestion.

Comment 5: The results section contains a lot of conclusions/discussion. The table and figures are not well explained.

Response 5: Thank you for the reviewer’s comment. we rewritten the discussion section according to the reviewer’s suggestion by eliminating the unnecessary details.

Comment 6:Are all of the cited references relevant to the research? This includes lots of references that are not relevant for this research.

Response 6: Thank you for the reviewer’s comment. We have been revised the manuscript by citing new references related to this field of study. 

Comment 7:Does this article provide a relevant contribution to the scientific discussion of this topic? This is not written well enough to determine if it provides a relevant contribution.

Response 7: Thank you for the reviewer’s comment. we rewritten the discussion section according to the reviewer’s suggestion by eliminating the unnecessary details.

Comment 8

Is it necessary to include study limitations in the discussion? Discussion needs to be relevant to the data.

Response 8: Thank you for the reviewer’s suggestion. We have been modified the revised manuscript according’s to reviewers’ suggestion.

Comment 9: Abstract does not present any results.

Response 9: Thank you for the reviewer’s comment. We have been modified the abstract of the revised manuscript according to the reviewer’s comments.

Comment 10: Line 44 - change 'mushrooms' to a specific class of fungi and provide a reference for the statement.

Response 10: Thank you for the reviewer’s suggestion. We have been modified the revised manuscript according’s to reviewers’ suggestion.

Comment 11: Line 66 - change to 'Many endophytic microorganisms that peacefully coexist...'

Response 11: Thank you for the reviewer’s suggestion. We have been modified the revised manuscript according’s to the reviewers’ suggestion.

Comment 12: Lines 80-92 - delete - not relevant to the paper or move to conclusions

Response 12: Thank you for the reviewer’s suggestion. We have been modified the revised manuscript by deleting the suggested sections.

Comment 13: Lines 106-110 - move to conclusions

Response 13: Thank you for the reviewer’s suggestion. We have been moved the suggested sentence to the conclusion part and modified the revised manuscript.

Comment 14: Line 113 - Need to provide identity of the fungi here. How were they cultured? How were they identified? How were they shown to be endophytes? Why were these fungi selected for the work presented here?

Response 14: Thank you for the reviewer’s comment. There are similar works associated with the study of the red algae Pterocladiella capillacea, in the previous literatures (37). We have been modified the revised manuscript by adding the section in materials and methods 2.1. The section rewritten as follows:

 ‘The isolation and identification of these fungi associated with macroalgae also been presented in previously published reports. Sterilized/washed Algal segments, from both healthy and deceased thalli were cultured glucose-yeast extract-peptone seawater agar, resulting in the isolation of pure fungal cultures. The identification process primarily relied on BLAST search analysis of the internal transcribed spacers of ribosomal DNA (ITS)’

We are sorry for any misunderstanding leading to the endophytes. The fungi used in this study are fungi associated with algae, which can be either epiphytes or endophytes.

We specifically chose six fungi because our previous study revealed that many cultured fungi associated with the red algae are known producers of antimicrobial secondary metabolites. In pursuit of efficient immunostimulants for shrimp aquaculture, this preliminary study aims to identify suitable candidates with biological activities.

Comment 15: Line 127 - change to '...GYSP medium, which consists of ...'

Response 15: Thank you for the reviewer’s suggestion. We have been modified the revised manuscript according’s to reviewers’ suggestion.

Comment 16: Line 164 - give concentration of sodium carbonate used

Response 16: Thank you for the reviewer’s comment. We are newly added the concentration details of sodium carbonate in the section 2.4 as 7.5% w/v

Comment 17: Line 192-194 - italicize the bacterial names2

Response 17: Thank you for the reviewer’s suggestion. We have been modified the revised manuscript according’s to reviewers’ suggestion.

Comment 18:Line 196-198 - Why was fungal testing included in the antibacterial section? No fungi were listed for testing.

Response 18: Thank you for the reviewer’s comment. Sorry for the typo. The section rewritten as follows:

The assay involved impregnating 6-mm-diameter filter paper disks with 100 µL of fungal filtrate extracts. These loaded disks were strategically placed on LB agar medium for antibacterial. Subsequently, the plates were incubated at 28°C for 3 days to promote fungal growth and at 37°C for 24 h to facilitate bacterial growth.

Comment 19: Line 211-213 - italicize the scientific name for brine shrimp

Response 19: Thank you for the reviewer’s suggestion. We have been modified the revised manuscript according’s to reviewers’ suggestion.

Comment 20: Line 222 - need to explain the positive control

Response 20: Thank you for the reviewer’s comment. We have been modified the Brine shrimp Assay section by adding the details of the related study.

Comment 21: Line 261 - section 3.1 - delete section/photos - no morphological data was presented

Response 21: Thank you for the reviewer’s comment. We have been modified the revised manuscript by deleting the specific section suggested by the reviewer.

Comment 22: Section 3.2 - change title to 'Antioxidant and Phytochemical Analyses'

- need to link fungal names with identifiers used in materials section

Response 22: Thank you for the reviewer’s suggestion. We have been modified the revised manuscript according’s to reviewers’ suggestion.

Comment 23: Line 273 - need to write out Trichoderma

Response 23: Thank you for the reviewer’s suggestion. We have been modified the revised manuscript by adding Trichoderma harzianum.

Comment 24: Line 276 - delete sentence '..reaffirming....capability.' This is a conclusion.

Response 24: Thank you for the reviewer’s suggestion. We have been modified the revised manuscript by deleting the suggested sentence.

Comment 25: Table 1 - write out genus name and add identifiers

Response 25: Thank you for the reviewer’s comment. Genus names and identifiers are updated in the revised manuscript.

Comment 26: Line 289 - change to 'No significant differences were noted.' or add letters to the table

Response 26: we have been modified the sentence “” to 'No significant differences were noted’. And updated the revised manuscript.

Comment 27: Line 279 - not what the table shows

Response 27: Sorry for the typo. We have been modified the manuscript according to the reviewer’s comment.

Comment 28: Line 281 - Table 1 don't match - Aspergillus sp. 7 -vs- Aspergillus niger - change one of these

Response 29: Thank you for the reviewer’s comment. We have modified the species name to Aspergillus sp. 7

Comment 29: Line 294-301 - move to methods section

Response 29: Thank you for the reviewer’s comment. We are revised the manuscript according to the reviewer’s suggestion.

Comment 30: Figure 2 - Is the survival rate compared to no extract added? If yes, write that in the legend.

Response 30: Thank you for the reviewer’s comment. We have added the control group for the study and updated the revised manuscript accordingly.

Comment 31: Line 315-317 - delete or move to conclusions

Response 31: Thank you for the reviewer’s comment. We have been modified the conclusion and revised manuscript accordingly.

Comment 32: Table 2 - Write out all genus names

Response 32: Thank you for the reviewer’s comment. Genus names are updated in the revised manuscript.

Comment 33: Table 3 - move to supplemental - need to explain column headings in legend

Response 5: Thank you for the reviewer’s suggestion. We have updated the revised manuscript by moving the Table 3 to the supplementary information.

Comment 34: : Delete figure 4 or move to supplemental

Response 34: Thank you for the reviewer’s suggestion.

We have modified the revised manuscript by moving the figure 4 to the supplementary information.

Comment 35:Lines 340-351 - Only include three important types of compounds.

Response 35: Thank you for the reviewer’s comment. At this point the explanation of all the others compound’s characteristic details is out of the scope for the study. We will try to include the detailed description in the future studies.

Comment 36Lines 356-366 - move to discussion

Response 36: Thank you for the reviewer’s comment. The manuscript has been modified according to reviewers’ suggestion.

Comment 37: Lines 381-440 - move to discussion

Response 37: Thank you for the reviewer’s comment. The manuscript has been modified according to the reviewers’ suggestions.

Round 2

Reviewer 1 Report

The manuscript discusses the significant findings of our research on marine fungi  associated with red seaweeds, particularly in relation to their antioxidant potential and metabolite analysis. In addition, In my opinion the authors responded exhaustively to my questions. So, if the Editor and other reviewers agrees, the revised manuscript may be accepted .

The revised manuscript has increased its scientific value. The figures and tables are correct. Appropriate statistical analysis was carried out.

Author Response

Response to Reviewer 1 Comments

Response 1: We are happy to hear that we revised the manuscript according to the reviewer’s suggestions.

Reviewer 2 Report

The corrections fixed the majority of problems.

Line 50 - Reference 7 is to a paper from 2018. That is not the 'present study'. Please fix the statement.

Table 1 - need to define IC 50 in legend.

Need to explain Blank in the Figure 2 legend.

Author Response

Response to Reviewer 2 Comments

Comment 1:Line 50 - Reference 7 is to a paper from 2018. That is not the 'present study'. Please fix the statement.

Response 1: Thank you for the reviewer's comment. We modified the typo in the sentence: “It was found in the present study that the three bioactive subfractions of low molecular weight secondary metabolites: ex-LMSI, ex-LMSII, and ex-LMSIII isolated from Cerrena unicolor culture fluid possessed anticancer, antioxidant, and antibacterial properties" To “In a recent study, it was reported that the three bioactive subfractions of low molecular weight secondary metabolites: ex-LMSI, ex-LMSII, and ex-LMSIII, isolated from Cerrena unicolor culture fluid, possessed anticancer, antioxidant, and antibacterial properties”

Comments 2: Table 1 - need to define IC 50 in legend.

Response 2: Thank you for the reviewer's comment. We modified the legend of the table by incorporating the term IC50 and modified the table 1 caption to: IC50 values of the Antioxidant activities (DPPH, ABTS) and Total phenolic, flavonoid, and tannin content of six marine fungi associated with the red algae P. capillacea.

Comments 2: Need to explain Blank in the Figure 2 legend:

Response 3 Thank you for the reviewer's comment. We are really sorry for the missing content in the caption. We rewrite the caption of Figure 2 by adding the missing term blank” and rewrite the caption as follows: “The cytotoxic effect of crude fungal extracts in comparison with blank (no extract added) on brine shrimp for 48 hours.”
